# Honor of Kings Arena: an Environment for Generalization in Competitive Reinforcement Learning

**Hua Wei**[*†♭]**, Jingxiao Chen**[*‡♭]**, Xiyang Ji**[*§]**, Hongyang Qin**[§]**, Minwen Deng**[§]**, Siqin Li**[§]**,**
**Liang Wang**[§]**, Weinan Zhang**[‡]**, Yong Yu**[‡]**, Lin Liu**[♮]**, Lanxiao Huang**[♮]**,**
**Deheng Ye**[§✉]**, Qiang Fu**[§]**, Wei Yang**[§]

[§]Tencent AI Lab, [♮]Tencent Timi Studio,
[†]New Jersey Institute of Technology, [‡]Shanghai Jiao Tong University
hua.wei@njit.edu, timemachine@sjtu.edu.cn, wnzhang@sjtu.edu.cn, yyu@apex.sjtu.edu.cn,
{xiyangji, hongyangqin, danierdeng, gracesqli, enginewang, lincliu, jackiehuang,
dericye, leonfu, willyang}@tencent.com

## Abstract

This paper introduces *Honor of Kings Arena*, a reinforcement learning (RL) environment based on *Honor of Kings*, one of the world's most popular games at present. Compared to other environments studied in most previous work, ours presents new generalization challenges for competitive reinforcement learning. It is a multi-agent problem with one agent competing against its opponent; and it requires the generalization ability as it has diverse targets to control and diverse opponents to compete with. We describe the observation, action, and reward specifications for the *Honor of Kings* domain and provide an open-source Python-based interface for communicating with the game engine. We provide twenty target heroes with a variety of tasks in *Honor of Kings Arena* and present initial baseline results for RL-based methods with feasible computing resources. Finally, we showcase the generalization challenges imposed by *Honor of Kings Arena* and possible remedies to the challenges. All of the software, including the environment-class, are publicly available at: `https://github.com/tencent-ailab/hok_env`. The documentation is available at: `https://aiarena.tencent.com/hok/doc/`.

## 1 Introduction

Games have been used as testbeds to measure AI capabilities in the past few decades, from backgammon [22] to chess [19] and Atari games [14]. In 2016, AlphaGo defeated the world champion through deep reinforcement learning and Monte Carlo tree search [19]. In recent years, reinforcement learning models have brought huge advancements in robot control [8], autonomous driving [16], and video games like StarCraft [23], Dota [1], Minecraft [7] and *Honor of Kings* [26, 28, 29].

Related to previous AI milestones, the research focus of game AI has shifted from board games to more complex games, such as imperfect information poker games [2] and real-time strategic games [28]. As a sub-genre of real-time strategic games, Multi-player Online Battle Arena (MOBA) games have attracted much attention recently [1, 29]. The unique playing mechanics of MOBA involve role/hero play and multi-player. Especially since MOBA games have different roles/heroes and each role has different actions, a good AI model needs to perform stably well in controlling the

---

[*] Authors contributed equally; [♭] work done at Tencent
[✉] Corresponding author

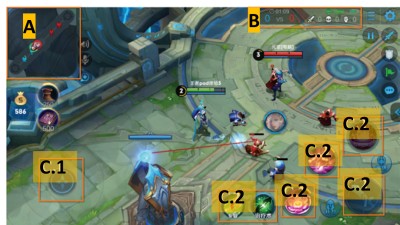 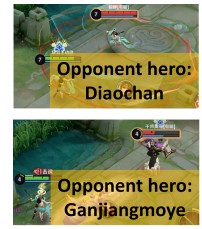 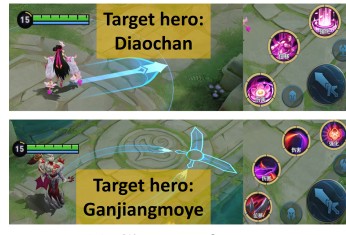

| (a) User interface | (b) Change of opponents | (c) Change of targets |

**Figure 1:** Game user interface (UI) and the change of opponents and targets in one match of *Honor of Kings*. (a) In the main screen, there are four sub-parts: a mini-map on the top-left, a dashboard that records the number of KDAs (kill/death/assist) on the top-right, a movement controller on the bottom-left, and skill controller buttons on the bottom-right. (b) The environment changes with different opponent heroes. (c) The action space changes with different target heroes.

actions of different heroes against different opponent heroes. This makes MOBA 1v1 games, which focus on hero control [29], a perfect testbed to test the generality of models under different tasks.

Existing benchmark environments on RL generality are mainly focusing on relatively narrow tasks for a single agent. For example, MetaWorld [30] and RLBench [10] present benchmarks of simulated manipulation tasks in a shared, table-top environment with a simulated arm, whose goal is to train the arm controller to complete tasks like opening the door, fetching balls, *etc*. As the agent's action space remains the same as an arm, it is hard to tell the generality of the learned RL on more diverse tasks like simulated legs.

In this paper, we provide *Honor of Kings Arena*, a MOBA 1v1 game environment, authorized by the original game *Honor of Kings*[1]. The game of *Honor of Kings* was reported to be one of the world's most popular and highest-grossing games of all time, as well as the most downloaded App worldwide. As of November 2020, the game was reported to have over 100 million daily active players [21]. There are two camps in MOBA 1v1, each with one agent, and each agent controls a hero character. As shown in Figure 1(a), an *Honor of Kings* player uses the bottom-left steer button to control the movements of a hero and uses the bottom-right set of buttons to control the hero's skills. To win a game, agents must take actions with planning, attacking, defending and skill combos, with consideration on the opponents in the partially observable environment.

Specifically, the *Honor of Kings Arena* imposes the following challenges regarding generalization:
• **Generalization across opponents**. When controlling one target hero, its opponent hero varies across different matches. There are over 20 possible opponent heroes in *Honor of Kings Arena* (in the original game there are over 100 heroes), each having different influences in the game environment. If we keep the same target hero and vary the opponent hero as in Figure 1(b), *Honor of Kings Arena* could be treated as a similar environment as MetaWorld [30], which both provides a variety of tasks for the same agent with the same action space.
• **Generalization across targets**. The generality challenge of RL arises to a different dimension when it comes to the competitive setting. In a match of MOBA game like *Honor of Kings* and DOTA, players also need to master different hero targets. Playing a different MOBA hero is like playing a different game since different heroes have various attacking and healing skills, and the action control can completely change from hero to hero, as shown in Figure 1(c). With over 20 heroes to control for *Honor of Kings Arena*, it calls for robust and generalized modeling in RL.

**Contributions:** As we will show in this paper, the above-mentioned challenges are not well solved by existing RL methods under *Honor of Kings Arena*. In summary, our contributions are as follows:
• We provide the *Honor of Kings Arena*, a highly-optimized game engine that simulates the popular MOBA game, *Honor of Kings*. It supports 20 heroes in the competitive mode.
• We introduce simple and standardized APIs to make RL in *Honor of Kings* straightforward: the complex observations and actions are defined in terms of low-resolution grids of features; configurable rewards are provided combining factors like the score from the game engine.
• We evaluate RL algorithms under *Honor of Kings Arena*, providing an extensive set of benchmarking results for future comparison.
• The generality challenges in competitive RL settings are proposed with preliminary experiments showing that existing RL methods cannot cope well under *Honor of Kings Arena*.

---

[1]https://en.wikipedia.org/wiki/Honor_of_Kings

**What *Honor of Kings Arena* isn't:** *Honor of Kings Arena* currently only supports the 1v1 mode of *Honor of Kings*, where there is only one hero in each camp. Though we acknowledge that some game modes of *Honor of Kings* (*i.e.*, 3v3 or 5v5 mode where there are multiple heroes in each camp) are more popular than 1v1 mode, they might complicates the generalization challenge in competitive RL with the cooperation ability between heroes (which is out of the scope of this paper). The cooperation ability also makes it hard to evaluate the generalization ability of different models. Thus, *Honor of Kings Arena* leaves these 3v3 and 5v5 mode of *Honor of Kings* out of the current implementation.

## 2 Motivations and Related Work

The key motivation behind *Honor of Kings Arena* is to address a variety of needs that existing environments lack:

**Diversity in controlled targets** A unique feature of *Honor of Kings Arena* is that it has 20 heroes for the agents to control, where each hero has its unique skills. In most existing open environments like Google Research Football [13], StarCraft2 AI Arena [17], Blood Bowl [12], MetaWorld [30] and RLBench [10], the meaning of actions remains the same when the agent controls different target units. The change of action control between a great number of heroes in *Honor of Kings Arena* provides appealing scenarios for testing the generalization ability of the agents. RoboSuite [31] provides seven different robotic arms across five single-agent tasks and three cooperative tasks between two arms. HoK environment differs from Robosuite by providing a large number of competitive settings.

**Free and open accessibility** DotA2 shares a similar game setting with *Honor of Kings* (both are representative MOBA games with large state/action spaces), which also has multi-agent competition and coordination, unknown environment model, partially observable environment and diverse target heroes to control, but the environment in [1] used by OpenAI is not open with only an overview posted. Other MOBA environments like Derk's Gym [15] lack free accessibility because of the requirement of commercial licenses. XLand [20] also focuses on the generalization capability of agents and supports multi-agent scenarios, but it is not open-source.

**Existing Interest** This environment has been used as a testbed for RL in research competitions [2] and many researchers have conducted experiments under the environment of Honor of Kings [3, 4, 11, 24, 25, 26, 28, 29, 27].Though some of them verified the feasibility of reinforcement learning in tackling the game [11, 26, 28, 29], they are more focused on methodological novelty in planning, tree-searching, *etc*. Unlike these papers, this paper focuses on making the environment open-accessible and providing benchmarking results, which could serve as a reference and foundation for future research. Moreover, this paper showed the weaknesses of former methods in lacking of model generalization across multiple heroes.

## 3 *Honor of Kings Arena* Environment

*Honor of Kings Arena* is open-sourced under Apache License V2.0 and accessible to all individuals for any non-commercial activity. The encrypted game engine and game replay tools follows Tencent's Hornor of Kings AI And Machine Learning License [3] and can be downloaded from: `https://aiarena.tencent.com/hok/download`. The code for agent training and evaluation is built with official authorization from *Honor of Kings* and is available at: `https://github.com/tencent-ailab/hok_env`. Any non-commercial users are free to download our game engine and tools after registration.

### 3.1 Tasks

We use the term "task" to refer to specific configurations of an environment (e.g., game setting, specific heroes, number of agents, etc.). The general task for agents in *Honor of Kings Arena* is as follows: When the match starts, each player controls the hero, sets out from the base, gains gold and experience by killing or destroying other game units (e.g., enemy heroes, creeps, turrets). The goal is to destroy the opponent's turrets and base crystal while protecting its own turrets and base crystal. A detailed description of the game units and heroes can be found in Appendix B.

---

[2]`https://aiarena.tencent.com/aiarena/en`
[3]`https://github.com/tencent-ailab/hok_env/blob/master/GAMECORE.LICENSE`

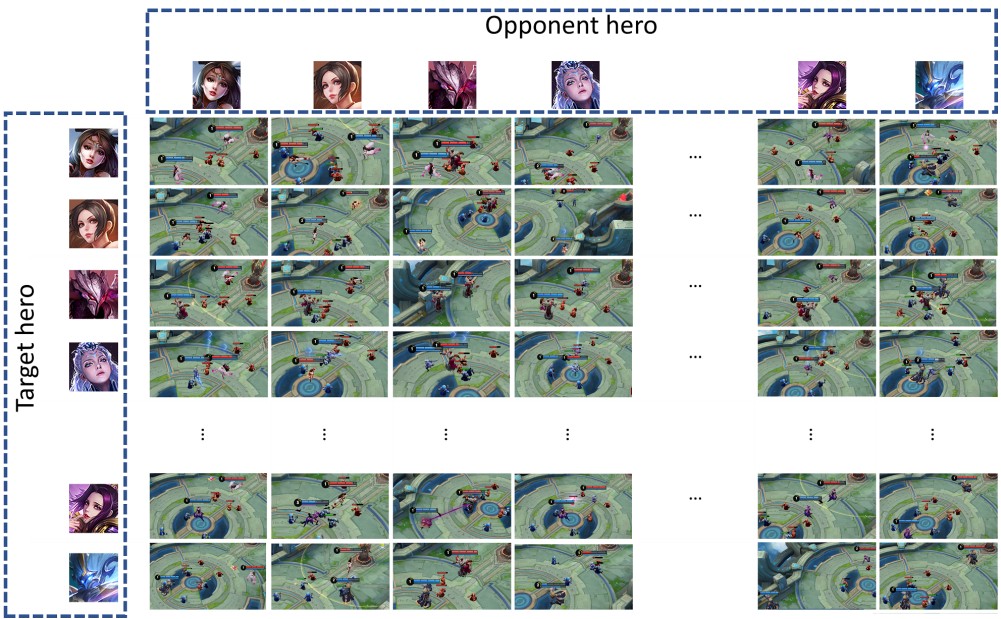

**Figure 2:** The tasks in *Honor of Kings Arena*. Each row represents the same target hero with different opponent heroes. Each column represents different target heroes with the same opponent hero. There are 20 heroes in *Honor of Kings Arena*, making $20 \times 20 = 400$ tasks in total.

Though the general goal is the same across different matches, every match would differentiate from each other. Before the match starts, each player needs to choose one hero to control, where each hero has its unique skills, which would have different influences on the environment. Any changes in the chosen hero would make the task different. As shown in Figure 2, in the current *Honor of Kings Arena*, 20 heroes could be chosen, which makes up 400 tasks in total.

### 3.2 Agents

*Honor of Kings Arena* provides recognizable and configurable observation spaces, action spaces, and reward functions. In this section, we provide a general description of these functions, whose details can be found in Appendix.

**Observation Space** The observations often carry spatial and status cues and suggest meaningful actions to perform in a given state. In *Honor of Kings Arena*, the observation space is designed to be the same across all heroes, creating the opportunity to generalize across tasks. Specifically, the observation space of *Honor of Kings Arena* consists of five main components, whose dimensions depend on the number of heroes in the game (for the full description, please see Appendix D): `HeroStatePublic`, which describes the hero's status; `HeroStatePrivate`, which includes the specific skill information for all the heroes in the game; `VecCreeps` describing the status of soldiers in the troops; `VecTurrets` describing the status of turrets and crystals; `VecCampsWholeInfo`, which indicates the period of the match.

**Action Space** The native action space of the environment consists of a triplet form, which covers all the possible actions of the hero hierarchically: 1) which action button to take; 2) who to target, *e.g.*, a turret, an enemy hero, or a soldier in the troop; 3) how to act, *e.g.*, the discretized direction to move and release skills. Note that different heroes have different prohibited skill offsets since they have different skills.

**Reward Information** *Honor of Kings* has both sparse and dense reward configurations in five categories: farming related, kill-death-assist (KDA) related, damage related, pushing related, and win-lose related (for the full description, please see Appendix F).

**Episode Dynamics** An episode of *Honor of Kings Arena* task terminates when the crystal of one camp is pushed down. In practice, there is a time limit in training, though an actual round of *Honor of Kings* game has no time limit. The timer is set at the beginning of the episode. The actions in *Honor of Kings Arena* are executed every 133ms by default to match with the response time of top-amateur players, while the action interval is configurable. The constraints of the game are expressed in system

state transitions of the game. For example, the HP of opponent's crystal will not decrease if the turret is not destroyed.

## 4    APIs and Implementation

**The RL Environment Class**    The class `hok1v1`, which can be found within the `HoK` module, defines two important functions `reset()` and `step()`, for starting a new episode, and for advancing time given an action, respectively. They both return a quadruple:

- `obs`: a list of NumPy arrays with the description of the agent's observation on the environment.

- `reward`: a list of scalars with the description of the immediate reward from the environment.

- `done`: a list of boolean values with the description of the game status.

- `info`: a list of `Dict` with length of agent number.

Specifically, each `Dict` in `info` describes the game information at current step and includes the following items:

- `observation` is a NumPy array including all six components mentioned in Section 3.2.

- `legal_action` describes current legal sub-actions with 1 NumPy array. The legal sub-actions incorporates prior knowledge of experienced human players and helps eliminate several unreasonable aspects: 1) skill or attack availability, e.g., the predicted action to release a skill within Cool Down time shall be eliminated; 2) being controlled by enemy hero skill or equipment effects; 3) hero-/item-specific restrictions.

- `sub_action_mask` is a NumPy array describing dependencies of different `Button` actions.

- `done` describes if the current game is over by a scalar value of 0 or 1.

- `frame_no` is the frame number of next state.

- `reward` is current reward value.

- `game_id, player_id` are string identifying the game and the run time ID of the current hero.

- `req_pb` is a string identifying protobuf information of the current hero.

We provide an example for using our environment in Listing 1.

**Technical Implementation and Performance**    The *Honor of Kings Arena* Engine is written in highly optimized C++ code, allowing it to be run on commodity machines both with GPU and without GPU-based rendering enabled. This allows it to obtain a performance of approximately 4.34 million samples (around 600 trajectories) per hour on a single 10-core machine that operates ten concurrent environments (see the subfigure in Figure 3). Furthermore, our infrastructure with sampler, memory pool and learner can easily scale over a cluster of machines. With 200 machines running in total 2000 concurrent environments, our training infrastructure allows over 800 million samples per hour.

```python
from hok import HoK1v1

# load environment
env = HoK1v1.load_game(game_config)

# init agents
agents = []
for i in range(env.num_agents):
    agents.append(Agent(agent_config))

# start an episode
obs, _, _, info  = env.reset()
total_reward = 0.0
done = False
while not done:
    actions = []
    for i in range(env.num_agents):
        action = agents[i].process(obs[i])
        actions.append(action)
        done = done or info[i]['done']
    obs, reward, done, info = env.step(actions)
```

**Listing 1:** Python example

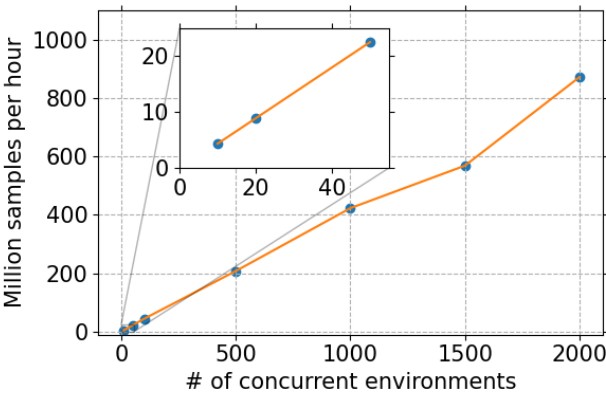

**Figure 3:** Number of steps per hour versus number of concurrent environments for the *Honor of Kings Arena* on a cluster of 10-core Intel Xeon Platinum 8255C CPUs with 2.50GHz. Our infrastructure could generate approximately 4.34 million samples per hour on a single 10-core machine, and could easily scale to larger concurrent environments.

**Competition and Evaluation Mechanism**   Similar to existing game environments, the agent has to interact with the environment and maximize its episodic reward by sequentially choosing suitable actions based on observations of the environment. The *Honor of Kings Arena* supports the competition across different kinds of agents: (1) Rule-based agents. A behavior-tree AI (*BT*) is integrated into *Honor of Kings* and provided with the environment. The *BT* uses rules that are provided by the game developers and handcrafted differently across different heroes to match the performance of a human player. [4] (2) Trained agents. The flexibility on competing against trained agents allows self-play schemes for training RL methods. All the baseline AIs, including the *BT* and trained models at different levels, are available. As we will show in Section 6, these different levels of models are important indicators for researchers to evaluate the performances from different methods.

## 5   Validation

The *Honor of Kings Arena* is an efficient and flexible learning environment. In order to validate the usefulness of *Honor of Kings Arena*, we conduct the evaluation with both heroes as Diaochan for training and implemented two baseline methods. In this scenario, the player needs to destroy one

---

[4]In *Honor of Kings* game, human players' levels are ranked as Bronze, Silver, Gold, Platinum, Diamond, Heavenly, King, and High King (from low to high). BT is hand authored by the game designers to match the Gold level, which can be treated as an entry level for humans in this paper.

turret before pushing down the opponent's crystal. In the following experiments, unless specified, we keep the heroes to control, the order of item purchasing, and skill upgrading the same for both camps to focus on learning tactics of the agents.

**Baselines** In the following experiments, we provide our validation results for existing deep reinforcement learning algorithms, including PPO [18] and Ape-X DQN [9]. We acknowledge that establishing baselines for reinforcement learning problems and algorithms could be notoriously difficult, therefore we refer directly to those that proved effective under the Honor of Kings environment [29, 25]. For a given algorithm, we ran experiments with similar network architecture, set of hyperparameters, and training configuration as described in the original papers.

**Feasibility under different resources** The first thing we need to validate is whether *Honor of Kings Arena* is able to support the training of RL under feasible computing resources. We run the policy over different CPU cores in parallel via self-play with mirrored policies to generate samples. We train the PPO network on one NVIDIA Tesla V100 SXM2 GPU. The results are summarized in Table 1. Table 1 shows that the convergence time with limited resources is less than 7 hours. As the number of CPUs for data collection increases, the training time to beat *BT* decreases.We also conducted experiments on multiple GPUs and found that it was the number of CPUs other than GPUs that impedes the training time to beat *BT*. With more CPU, the environment could generate more training samples, while with more GPU, the training sample could consume more frequently. When the consumption frequency, *i.e.*, number of times per sample used for training, is the same under different $\frac{\#CPUs}{\#GPUs}$, the training time to beat *BT* remains approximately the same.

**Table 1:** Training time to beat *BT* under different computing resources (standard deviation, each experiment run 3 times). With limited CPU cores, the self-play agent is able to beat *BT* using around 6 hours, and the time using additional CPU cores over 512 becomes about one hour.

| CPU Cores | Training hours | Consumption freq. |
|-----------|----------------|-------------------|
| 128       | 6.16±0.009     | 23.80 ±0.87       |
| 256       | 1.67±0.021     | 10.60±0.64        |
| 512       | 1.08±0.025     | 5.21±0.07         |
| 1204      | 0.90±0.06      | 2.69±0.03         |
| 2048      | 0.89±0.06      | 1.30±0.008        |

**Performance of Different Models** We provide experimental results for both PPO [29] and DQN [25] for the *Honor of Kings Arena* in Figure 4. The experimental results indicate that the *Honor of Kings Arena* provides challenges in the choice of RL methods toward a stronger AI, with 2000M samples used for the model of PPO to beat the *BT*. Although both PPO and DQN can beat *BT* within 3000M samples, they do not achieve the same final performance in terms of reward.

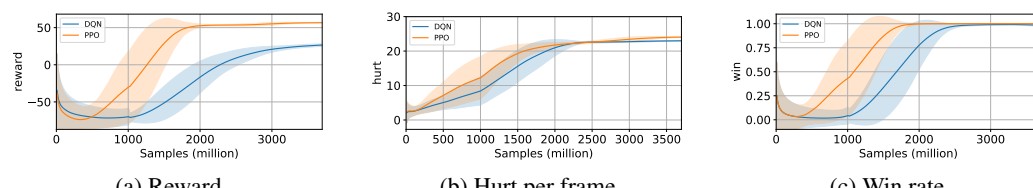

| (a) Reward | (b) Hurt per frame | (c) Win rate |
|:---:|:---:|:---:|

**Figure 4:** Different evaluation metrics on the *Honor of Kings Arena* for DQN and PPO w.r.t. the number of training samples. Error bars represent standard deviation. PPO performs better than DQN.

**Performance against *BT*** We present preliminary results showing the possibility of using RL to achieve human-level performance. We run experiments with over 1024 CPU cores and one GPU core via self-play. The Elo score [5] is used to evaluate the relative performance of one model among all the models. From Figure 5, we can see that PPO can beat *BT* (indicating the level of a normal human), within two hours and takes about 10 hours to converge.

## 6  Research Direction: Generalization Challenge in Competitive RL

In *Honor of Kings Arena*, players could control different heroes against different opponents, and the environment is also changed by the hero's actions. A policy with good transferability is expected to perform stably against different opponents. We conducted two experiments with PPO algorithm to showcase the transferability challenge.

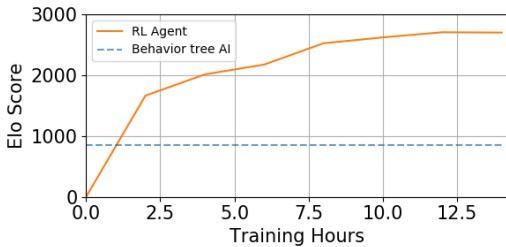

**Figure 5:** The Elo score of models under different training times. Blue lines shows the Elo score for $BT$. The self-play trained PPO could beat $BT$ in about 1 hour, and converge after 10 hours.

**Generalization Challenge across Opponents** In the first experiment, the policy is trained under the task of "Diaochan (RL) vs. Diaochan ($BT$)", and tested under the tasks of "Diaochan (RL) vs. different opponent heroes ($BT$)" for 98 rounds of matches. In these tasks, the RL model is only used to control Diaochan, and its opponents are changed as different tasks. The results are shown in Figure 6. The model trained on Diaochan could beat the opponent Diaochan controlled by $BT$ (90% winning rate) since the target to control during testing and training is the same. However, when the opponent hero changes, the performance of the same trained policy drops dramatically, as the change of opponent hero differs the testing setting from the training setting, indicating the lack of transferability of the policy learned by existing methods.

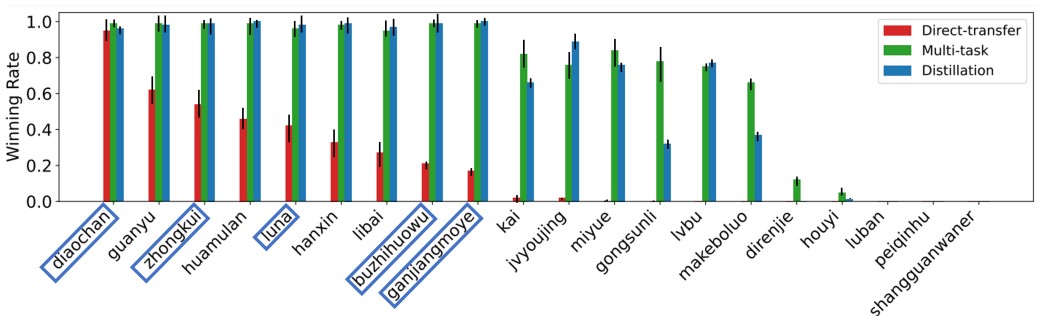

**Figure 6:** Win rate of a well-trained model from task "Diaochan (RL) vs. Diaochan ($BT$)" transferred to tasks "Diaochan (RL) vs. different opponent heroes ($BT$)". The agent is trained to control Diaochan against Diaochan controlled by $BT$, and tested to control Diaochan against different heroes controlled by $BT$. Red: Directly transferring the model to control Diaochan and compete with different opponent heroes. Green: Multi-task training on five tasks "Diaochan (RL) vs. Diaochan/Buzhihuowu/Luna/Ganjiangmoye/Zhongkui ($BT$)" and testing the model on twenty tasks. Blue: Distilling the model trained from five tasks "Diaochan (RL) vs. Diaochan/Buzhihuowu/Luna/Ganjiangmoye/Zhongkui ($BT$)" and testing the model on twenty tasks. The policy trained on Diaochan could not generalize to all tasks with different *opponent* heroes. Blue rectangles highlights the five tasks used in multi-task and distillation. The error bars indicate the standard deviation under five seeds.

**Generalization Challenge across Targets** In the second experiment, the policy is trained under the task of "Diaochan (RL) vs. Diaochan ($BT$)", and tested under the task of "different target heroes (RL) vs. Diaochan ($BT$)" for 98 rounds of matches. In these tasks, the RL model is used to control different heroes, but their opponents remains the same as Diaochan. The results are shown in Figure 7. When the target changes from Diaochan to other heroes, the performance of the same trained policy drops dramatically, as the change of target heroes differs the action meaning from the Diaochan's action in the training setting.

**Remedies: Multi-task and Distillation** A possible fix to enhance transferability in RL policies is to do multi-task training, *i.e.*, adding multiple testing settings during training, and forcing the training setting the same as the testing. We conducted an additional experiment to show that these methods show improvements on certain tasks. For multi-task models, we train the model with five tasks "Diaochan (RL) vs. Diaochan/Buzhihuowu/Luna/Ganjiangmoye/Zhongkui", and test it in twenty tasks. The results are shown in both Figure 6 and Figure 7. We can see that multi-task training improves the performance in all the test tasks.

Another possible enhancement for transferability is to distill one model from multiple models. We use the student-driven policy distillation that is proposed in [6]. We conducted an experiment to show that this method could achieve similar improvements to multi-task training over directly transferring

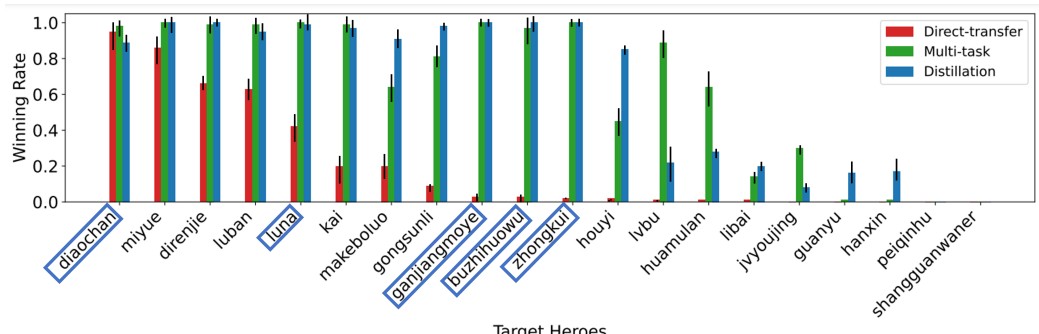

**Figure 7:** Win rate of a well-trained model from task "Diaochan (RL) vs. Diaochan (*BT*)" transferred to tasks "Different target heroes (RL) vs. Diaochan (*BT*)". The agent is trained to control Diaochan against Diaochan controlled by *BT*, and tested to control different heroes against Diaochan controlled by *BT*. Red: Directly transferring the model to control Diaochan and compete with different opponent heroes. Green: Multi-task training on five tasks "Diaochan/Buzhihuowu/Luna/Ganjiangmoye/Zhongkui (RL) vs. Diaochan (*BT*)" and testing the model on twenty tasks. Blue: Distilling the model trained from five tasks "Diaochan/Buzhihuowu/Luna/Ganjiangmoye/Zhongkui (RL) vs. Diaochan (*BT*)" and testing the model on twenty tasks. The policy trained on Diaochan could not generalize to all tasks with different *target* heroes. Blue rectangles highlights the five tasks used in multi-task and distillation. The error bars indicate the standard deviation under five seeds.

models. In both Figure 6 and Figure 7, the model is distilled from five models trained on five different tasks and tested in twenty tasks. We can see that the distillation improves the performance in all the test tasks.

In the experiments above, we only chose the task "Diaochan vs. Diaochan" as the primary training task. However, other feasible training tasks, such as "Buzhihuowu vs. Buzhihuowu", are reasonable options. We chose 5 different heroes as the primary training tasks and test generalization across both opponents and targets with direct transfer in Appendix H.

**Multi-level Models for Evaluation** Although multi-task and distillation could improve the generalization performance on many tasks, we also noticed that for certain tasks like opponent heroes changed to Peiqinhu/Shangguanwaner in Figure 6, or target heroes changed to Peiqinhu/Shangguanwaner in Figure 7, the winning rate for the Diaochan model is constantly zero. This makes it hard to evaluate the performance of different techniques for generalization.

Similarly, we noticed that both multi-task and distallation achieves near 100% winning rate in tasks like opponent heroes changed to Buzhihuowu *etc* in Figure 6, or target heroes changed to Buzhihuowu *etc* in Figure 7. To help researchers better evaluate their models, we also provided different levels of models for different heroes other than *BT*. One example of different levels of Buzhihuowu as the opponents for Diaochan is shown in Figure 8. If we only provide the level-3 model of Buzhihuowu, the winning rate would be tricky and hard to tell the actual performance of trained Diaochan model in different training hours.

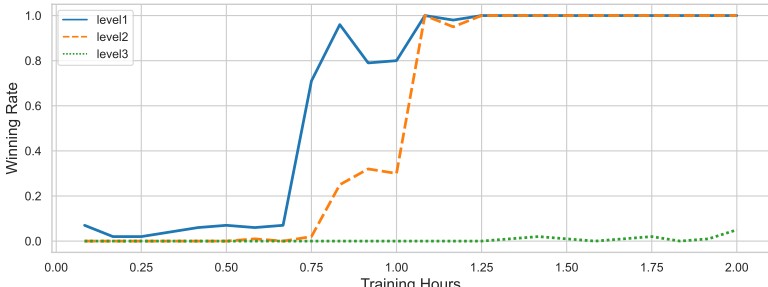

**Figure 8:** Winning rate of an RL agent for Diaochan to beat an opponent hero Buzhihuowu at different levels.

# 7 Conclusion and Future Work

The *Honor of Kings Arena* is a starting place for the design and performance comparison of competitive reinforcement learning algorithms with generalization necessity across different tasks. The configurable state and reward functions allow for easy-to-use suite-wide performance measures and explorations. The results presented in this paper constitute baselines using well-performing implementations of these algorithms.

We are excited to share the *Honor of Kings Arena* with the broader community and hope that it will be helpful. We look forward to the diverse research the environment may enable and integrating community contributions in future releases. In the future, we plan to optimize the deployment of our environment across different platforms. Meanwhile, we will also organize more competitions based on *Honor of Kings Arena* and provide our computing resources to contestants.

## Acknowledgement

The SJTU team is partially supported by "New Generation of AI 2030" Major Project (2018AAA0100900) and National Natural Science Foundation of China (62076161). The author Jingxiao Chen is supported by Wu Wen Jun Honorary Doctoral Scholarship, AI Institute, Shanghai Jiao Tong University.

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
