## A  License and Documentations

*Honor of Kings Arena* is open-sourced under Apache License V2.0. The code for agent training and evaluation is built with official authorization from *Honor of Kings* and is available at: `https://github.com/tencent-ailab/hok_env`, containing the game engine and the affiliated code for agent training and evaluation. The encrypted game engine and game replay tools can be downloaded from: `https://aiarena.tencent.com/hok/download`. All experiments can be reproduced from the source code, which includes all hyper-parameters and configuration. *Honor of Kings Arena* is authorized by the game developers of *Honor of Kings* and the authors will bear all responsibility in case of violation of rights, etc., ensure access to the data and provide the necessary maintenance. We also provide the detailed documentations for *Honor of Kings Arena* and *Honor of Kings* in this link: `https://aiarena.tencent.com/hok/doc/`, including the following:

- Quick start of *Honor of Kings Arena*
    - Installation
    - First agent
    - Evaluation
- Description of *Honor of Kings Arena*
    - Game Description of *Honor of Kings*
    - Observation space
    - Action space
    - Reward
- Game Description of *Honor of Kings*
- *Honor of Kings Arena* package API Reference

## B  Basic Units

**Hero**  Heroes are player-controlled units that can move around and have abilities to release various attacking and healing skills. There are many heroes with different attributes, skills and tactics in the full game settings for *Honor of Kings*. Each player can choose one hero to control and team up with others as a lineup of heroes. For more information on hero skills, please refer to the official website of *Honor of Kings*: `https://pvp.qq.com/m/m201706/heroList.shtml`. As is shown in Figure 9, *Honor of Kings Arena* now supports 20 heroes, because these 20 heroes are sufficient for testing the generalization ability. We will support more heroes in the future.

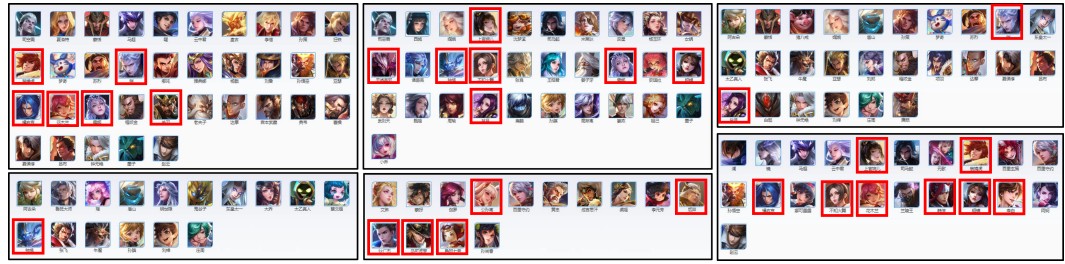

**Figure 9:** Subset of the *Honor of Kings* heroes. There are six general types of heroes and each hero has a specific set of hero attributes and skills. Red rectangles highlight the heroes provided in current *Honor of Kings Arena*.

**Creep**  Creeps are a small group of computer-controlled creatures that periodically travel along the predefined lane to attack the opponent units.

**Turret**  Turrets, i.e., defensive buildings, are designed to attack any opponent units moving into their sight area.

**Crystal**    The crystal, located in the base of each player, can attack opponent units around their sight area with higher damages than turrets. The final victory condition is to push down the enemy's crystal. And the opponent units cannot cause damage to the base towers without destroying the turrets.

# C    Hero Details

## C.1    Basic Attributes

The basic attributes are what every hero has: health point (HP) volume and magical points (MP) volume, attack, defense, resistance, *etc*.

**Table 2:** Basic attributes of a hero

| | |
|---|---|
| Maximum HP | Maximum MP |
| Physical attack | Magical attack |
| Physical bloodsucking | Magical bloodsucking |
| Physical defense | Magical defense |
| Physical penetration | Magical penetration |
| HP regeneration speed | MP regeneration speed |
| Attack speed | Cool-down time |
| Bonus attack chance | Resilience |
| Attack range | Movement speed |

Among all the basic attributes, HP and MP are essential for the states of the hero, where the hero would die when HP runs out and the hero would not be able to release magical skills if MP runs out.

Defense and attack exist in pairs. There are three types of attacks: physical damage, magic damage, and true damage. Among them, physical defense is only effective for physical damage, magical defense is only effective for magical damage, and true damage is effective regardless of physical and magical defense.

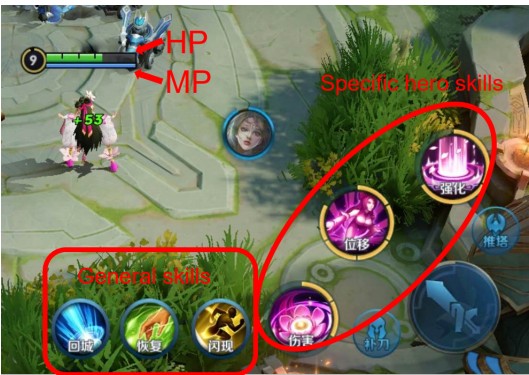

**Figure 10:** Important basic attributes of the hero like HP and MP are shown in the UI of *Honor of Kings*, together with the state of hero skills.

## C.2    Skills

**General skills**    General skills are independent skills players need to choose according to their hero type before the game starts. The general skill does not require any consumption of experiences or gold. Once used by the hero, it will be available only after a certain cooling-down time.

**Specific hero skills**    Each hero has a specific set of hero skills, which is different from all other heroes. Each skill has different levels, which can be upgraded with experiences from killing or clearing opponent units. It usually consumes MP to release specific hero skills.

### C.3 Hero Types

Generally, there are six types of heroes with different playing strategies, including:

- Tank. Tank-type heroes have high blood and defense, and are heroes with strong survivability and average damage ability. They usually occupy the front row position in team battles, resist the damages from enemies.

- Warrior. Warrior-type heroes have balanced offensive and defensive capabilities. They usually stand behind tank heroes in team battles, take a small amount of enemy damage, fight in the enemy lineup, and sometimes act as a vanguard in the absence of tank heroes and take a lot of damage.

- Assassin. Assassin-type heroes are heroes with weak survivability, but extremely explosive damage capabilities. In a team battle, the assassin can go to the back of the enemy lineup and find the right time for a set of skills to kill the enemy heroes with low blood.

- Mage. Mage-type heroes are heroes with weak survivability, but high magical damage and control skills. They usually occupy the back-row position in the team battle and try to control the enemy heroes.

- Marksman. Marksman-type heroes are heroes with extremely high remote physical damage capabilities and control skills. In team battles, Marksman can use their remote damage advantage to stand in the back row.

- Support. Support heroes are heroes with mediocre survivability and output ability. They rely on powerful skills to increase the status of teammates and control enemy heroes.

### C.4 Items

In *Honor of Kings*, heroes can not only upgrade to higher-level skills with experiences but also can purchase items with golds. Each hero is allowed to possess up to six items. The types of items in *Honor of Kings* are various and complex. When choosing items, the players need to decide which items to buy, keep and sell to strengthen the hero's characteristics. For example, tank-type heroes might need shields to enhance defense, warriors and mages might need to enhance their damage, *etc*. However, the choices of items are also influenced by the opponents and their items during different phases of the game. For example, Suppose the opponent is a mage that relies on magical attacks. In that case, the player can choose magical resistance items, making it difficult for the opponent to do more magical damage.

## D State Vector

Users can customize and redefine their own observations from the returned 'info' from 'env.step()', while we provide a basic set of observations here. The observation space of *Honor of Kings Arena* consists of five main components, whose dimensions depend on the number of heroes in the game: The most notable component is `HeroStatePublic`, which describes the hero's status, including whether it's alive, its ID, its health points (HP), and its skill status, and its skill related information, *etc*. `HeroStatePrivate` includes the specific kill information for all the heroes in the game. `VecCreeps` describes the status of soldiers in the troops, including their locations, HPs, *etc*. `VecTurrets` describes the status of turrets and crystals. It should be noted that the states of enemy units, even if they're invisible to the ego camp, are also available, which could be helpful for algorithms that learn from hindsight and should be masked out during policy execution. The last part of the observation space is `VecCampsWholeInfo`, which indicates the period of the match. Their dimensions and descriptions can be found in Table 3. The meanings of every dimension in the state space can be found in the link provided in Documentation of Section A.

## E Action Space

The native action space of the environment consists of a triplet form, *i.e.*, the action button, the movement offset over the x-axis and y-axis, the skill offset over the x-axis and y-axis, and the target game unit. It covers all the possible actions of the hero hierarchically: 1) what action button to take; 2) who to target, e.g., a turret, an enemy hero, or a soldier in the troop; 3) how to act, e.g., the

**Table 3:** State information

| Feature Class | Field | Description | Dimension |
|---|---|---|---|
| `HeroStatePublic` | Hero Status | HP, MP, level, exp, position, skills, *etc* | 49 * 2 |
| | Hero skills | Specific attributes of hero skills, like buff mark layers, *etc* | 53 * 2 |
| `HeroStatePrivate` | Diaochan-related | skill position and buffs | 11 |
| | Luna-related | phases and status of normal attack | 7 |
| | Jvyoujing-related | strengthened normal attack related | 9 |
| | Luban-related | attack stage | 5 |
| `VecCreeps` | Status | HP, camp, attack range, etc. | 12 * 4 |
| | Position | Absolute and relative coordinates, distance | 6 * 4 |
| `VecTurrets` | Status | Hp, camp, attack range, etc. | 12 * 4 |
| | Position | Absolute and relative coordinates, distance | 6 * 4 |
| `VecCampsWholeInfo` | Current Period | Divide game time into 5 periods | 5 |
| Total | Sum | All features | 491 |

discretized direction to move and release skills. As illustrated in Figure 11, the action buttons include move, attack, skill releasing, etc. The details of the action space, including the types and meanings of every dimension, can be found in Table 4.

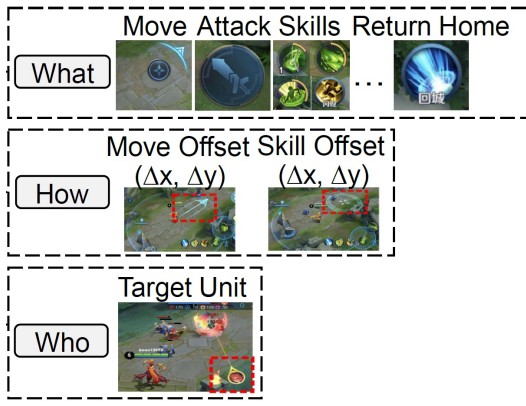

**Figure 11:** The action space is composed of action button, offsets and targets.

## F   Reward Information

Users can customize and redefine their own rewards (including the termination rewards) from the returned 'info' from 'env.step()', while we provide a basic set of rewards here. *Honor of Kings* has both sparse and dense reward configurations in five categories: 1) Farming related: the amount of gold and experience, and the penalty of not acting, which are dense reward signals; 2) KDA related: the number of kill, death and assist, and the last hit to enemy units, which are sparse reward signals; 3) Damage related: a dense reward - the number of health point, and a sparse reward - the amount of attack to enemy hero; 4) Pushing related: the amount of attack to enemy turrets and crystal, which are dense rewards; 5) Win/lose related: destroy the enemy home base, which are sparse reward signals received at the end of the game.

It is possible that only using the dense reward version of the *Honor of Kings* will likely resemble the sparsity, which is often seen in previously sparse rewarding benchmarks. Given the sparse-reward nature of this task, we encourage researchers to develop novel intrinsic reward-based systems, such as curiosity, empowerment, or other signals to augment the external reward signal provided by the environment.

**Table 4:** Action Space description

| Action Class | Type | Description | Dimension |
|---|---|---|---|
| Button | None | No action | 1 |
| | Move | Move hero | 1 |
| | Normal Attack | Release normal attack | 1 |
| | Skill 1 | Release skill 1st | 1 |
| | Skill 2 | Release skill 2nd | 1 |
| | Skill 3 | Release skill 3rd | 1 |
| | Heal Skill | Release heal skill | 1 |
| | Summoner Skill | Release summoner skill | 1 |
| | Recall | Start channeling and return to home crystal after a few seconds if not interrupted | 1 |
| | Skill 4 | Release skill 4th (Only valid for certain heroes) | 1 |
| | Equipment Skill | Release skill provided by certain equipment | 1 |
| Move | Move X | Move direction along X-axis | 16 |
| | Move Z | Move direction along Z-axis | 16 |
| Skill | Skill X | Skill direction along X-axis | 16 |
| | Skill Z | Skill direction along Z-axis | 16 |
| Target | None | Empty target | 1 |
| | Self | Self player | 1 |
| | Enemy | Enemy player | 1 |
| | Soldier | 4 Nearest soldiers | 4 |
| | Tower | Nearest tower | 1 |

**Table 5:** Reward Design

| Reward | Weight | Type | Description |
|---|---|---|---|
| hp_point | 2.0 | dense | the rate of health point of hero |
| tower_hp_point | 10.0 | dense | the rate of health point of tower |
| money (gold) | 0.006 | dense | the total gold gained |
| ep_rate | 0.75 | dense | the rate of mana point |
| death | -1.0 | sparse | being killed |
| kill | -0.6 | sparse | killing an enemy hero |
| exp | 0.006 | dense | the experience gained |

# G   Hyperparameters

The network structure is directly adopted from [29] whose implementation can be found in our code `https://github.com/tencent-ailab/hok_env/blob/main/code/common/algorithm.py`. We use Adam optimizer with initial learning rate 0.0001. For PPO, the two clipping hyperparameters $\epsilon$ and $c$ are set as 0.2 and 3, respectively. The discount factor is set as 0.997. For the case of Honor of Kings, this discount is valuing future rewards with a half-life of about 46 seconds. We set $\lambda = 0.95$ in GAE to reduce the variance caused by delayed effects. For DQN, the discount factor of tagert Q-network is set as 0.98.

# H   Additional Experiments

The *Honor of Kings Arena* provides scenarios and flexible APIs to train different reinforcement learning models. When training deep RL models, it often requires crafting designs like network structure and loss function based on different problems. In the following sections, if not specified, we will use Diaochan vs. Diaochan as our task in the experiment.

## H.1   Dual clip in PPO

To tackle the dramatic shift between different rounds of data in off-policy reinforcement learning, one common design to apply clipping, on either policy entropy or trajectory importance ratios [18].

In [28], it utilized a dual-clip PPO to support large-scale distributed training. And we investigate the influences of this design and found out the model with dual clip is slightly better than the original PPO model, as is shown in Figure 12.

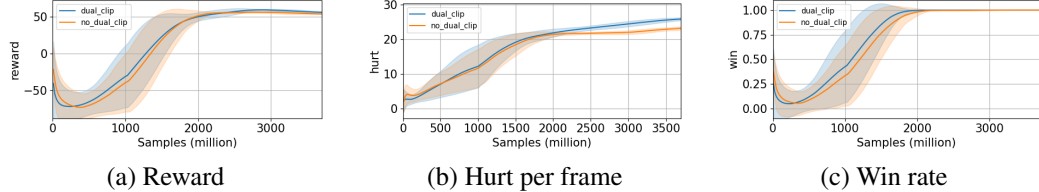

(a) Reward          (b) Hurt per frame          (c) Win rate

**Figure 12:** Different evaluation metrics on the *Honor of Kings Arena* 1v1 mode for the ablation study on dual-clip w.r.t. the number of training samples. Error bars represent standard deviation.

## H.2 Legal action mask

To improve the training efficiency and incorporate the prior knowledge of human players, especially for RL problems with large action spaces. In *Honor of Kings Arena*, it provides legal action information to help eliminate unreasonable actions for reference, where researchers can design their own legal actions based on their knowledge. We can use the legal action information to mask unreasonable actions out during the training process. The experiment results is shown in Figure 13. As expected, without legal action, the agent quickly converges to a local optimum under the large action space, and it is critical to use action mask for better performances.

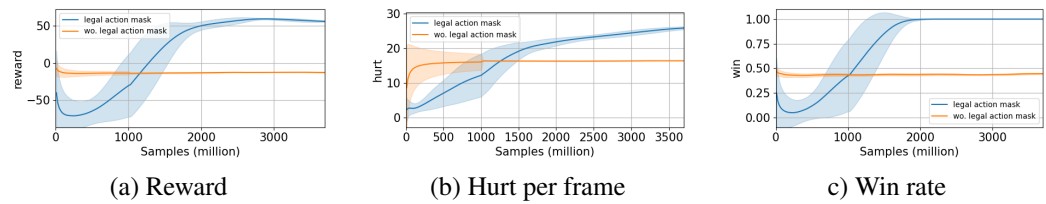

(a) Reward          (b) Hurt per frame          c) Win rate

**Figure 13:** Different evaluation metrics on the *Honor of Kings Arena* 1v1 mode for the ablation study on legal action mask. Error bars represent standard deviation.

## H.3 LSTM for partial observation

To tackle the challenge from the partially observable environment, researchers often utilize LSTMs in the model network. In *Honor of Kings Arena*, the action sequences of a hero, also called skill combos, are critical to creating severe and instant damage and should be incorporated with temporal information. We compared the models w/wo LSTMs, and the results are shown in Figure 14. As expected, the performance of models with LSTMs performs better than the model without LSTMs.

## H.4 Evaluation under Elo

When a policy is trained against a fixed opponent, it may exploit its particular weaknesses and may not generalize well to other adversaries. We conducted an experiment to showcase this in which a first model A was trained against *BT*. Then, another agent B was trained against a frozen version of agent A in the same scenario. While B managed to beat A consistently, its performance against built-in AI was not as good as Model A. The numerical results showing this lack of transitivity across the agents are presented in Table 6.

To evaluate the ability of different agents, we suggest the necessity of using Elo score. We evaluate the Elo score for all the models, and the K factor in Elo score calculation is set to 200. As is shown in Table 6, although Model B can perfectly beat Model A, its Elo score is less than A's. This is because because B is trained against a *frozen* version of A and might overfit to A's weaknesses. The deviation of the winning rate (0.25) in B *vs. BT* is much larger than A *vs. BT* (0.03), which also indicates that B is worse than it shows in the mean of winning rates (0.91). This shows that Elo score considers the

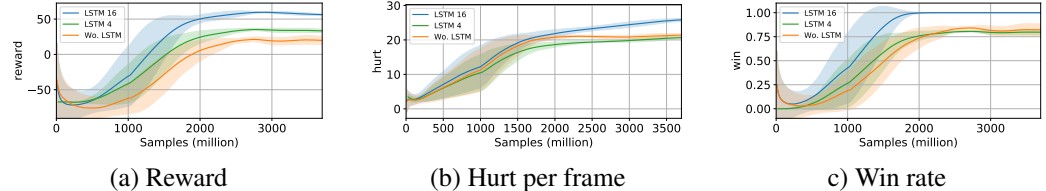

| (a) Reward | (b) Hurt per frame | c) Win rate |

**Figure 14:** Different evaluation metrics on the *Honor of Kings Arena* 1v1 mode for the ablation study on LSTM w.r.t. the number of training samples. Error bars represent standard deviation. With LSTMs, the model performs better against models without LSTMs; as the step size grows, the model performs better.

**Table 6:** Performances of different agents under the evaluation of winning rate (with standard deviation) and Elo score. Although B could beat A among all the matches, B's winning rate is not as good as A's when competing with$BT$. Instead, the Elo scores can reflect the relative capability.

| Models in matches | Winning rate |
|---|---|
| Model A vs. $BT$ | 0.91±0.03 |
| Model B vs. Model A | 1.00±0.00 |
| Model B vs. $BT$ | 0.81±0.25 |

| Models | Elo Score |
|---|---|
| Model A | 2372 |
| Model B | 1186 |
| $BT$ | 0 |

overall performance of one model against all the models, which is a more comprehensive metric than measuring the winning rate of one model against another model.

### H.5   Tasks on different heroes

In Section 6, we chose the task "Diaochan vs. Diaochan" as the primary training task, while other feasible training tasks are reasonable options. Here we provide 5 different heroes as the primary training tasks and test their generalization ability across both opponents and targets with direct transfer. The results are shown in Figure 15 and Figure 16. The corresponding bar charts are provided as well in Figure 17 and Figure 18.

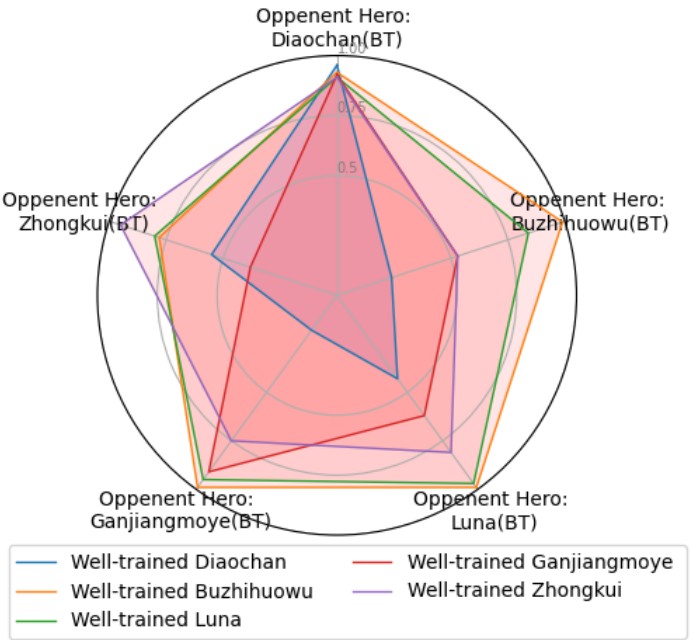

**Figure 15:** Generalization across different opponents with models trained from different heroes. Each line represents a well-trained model trained to control a target hero to compete with the same hero controlled by $BT$. Each axis in the plot represents the testing task of different opponent hero to compete with for the same well-trained model. The value on the axis represents the winning rate under the task. A model trained on Luna and Buzhihuowu can generalize better to compete with four other heroes controlled by $BT$.

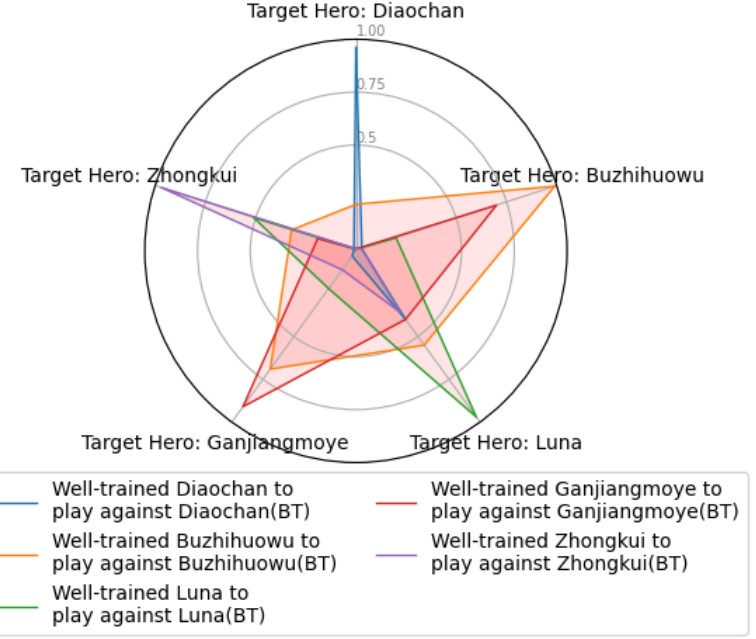

**Figure 16:** Generalization across different target heroes with models trained from different heroes. Each line represents a well-trained model trained to control a target hero to compete with the same hero controlled by $BT$. Each axis in the plot represents the testing task of different target hero to control for the same well-trained model. The value on the axis represents the winning rate under the task. A model trained on Buzhihuowu can generalize better than models trained on other heroes to control different heroes.

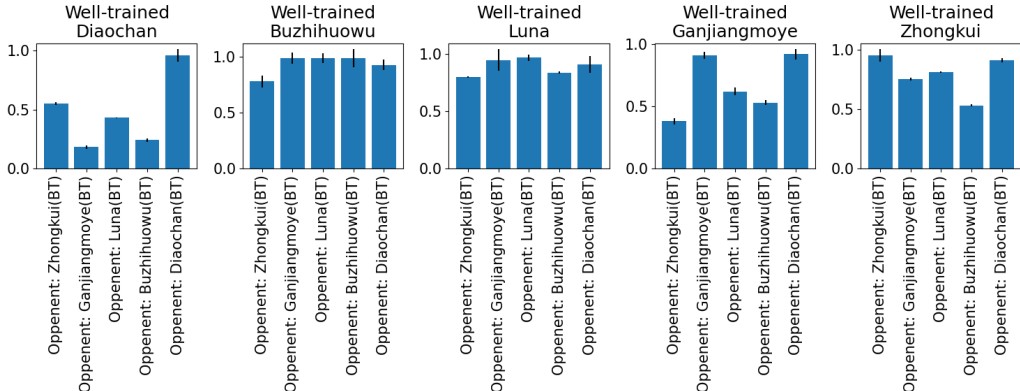

**Figure 17:** Bar charts of Figure 15.

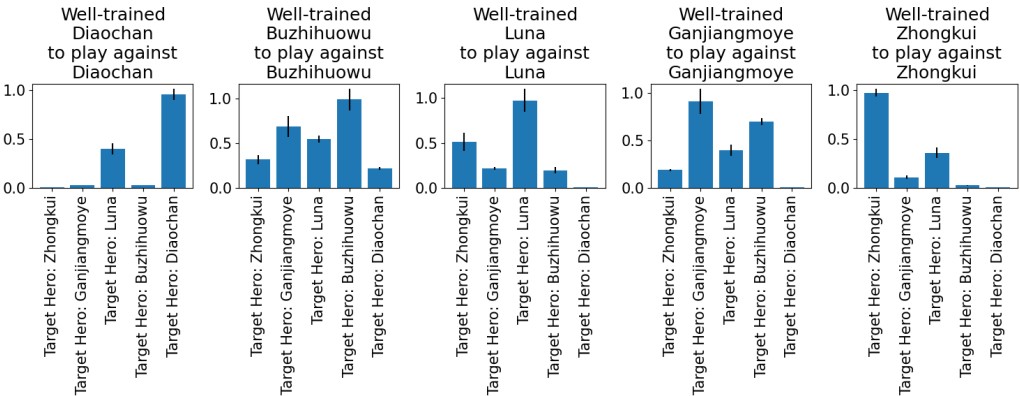

**Figure 18:** Bar charts of Figure 16.