# OpenReview forum: "Honor of Kings Arena: an Environment for Generalization in Competitive Reinforcement Learning"
_NeurIPS.cc/2022/Track/Datasets_and_Benchmarks — NeurIPS 2022 Datasets and Benchmarks _

### Official Review · Reviewer_eoxv · 2022-06-30
**A Review**

**Rating:** 7
**Confidence:** 2
**Correctness:** Yes.
**Clarity:** Yes.

**Strengths:**

Dataset appears to be useful for future research.

**Weaknesses:**

Isn't exactly innovative. Other sources for RL data generation are available.

Generalization across heroes is not particularly innovative as starcraft 2 provides generalization across races.

Missed potential for multi agent team based RL.



**Additional Feedback:**

Consider 3v3 or other formats to increase the innovativeness of the paper.

**Documentation:**

Yep. It appears to be properly documented. No DOI is provided, but it is a data generator.

**Ethics:**

No concerns.

**Relation To Prior Work:**

Does not significantly extend the work done by The StarCraft Multi-Agent Challenge.

**Summary And Contributions:**

Reasonably useful dataset for RL. Unfortunately, I am not an expert in this field, but the author seems to provide suitable tools to efficiently explore the action space.

Benchmarks demonstrate that basic hardware is capable of utilizing this data generator and training a RL model.

Generalization across heroes can be of interest for transfer learning.

---

### Official Review · Reviewer_GSzC · 2022-07-03

**Rating:** 7
**Confidence:** 4

**Strengths:**

**"Real world" game simulator**. The game provided is a popular and highly competitive game. Prior work has not provided access to this level of game complexity, instead focusing on more "toy" tasks. Contributing access to this type of game can spur further research in RL in general for these types of environments, including the generalization topic the paper emphasizes. While practitioners have worked on these domains in partnership with companies (OpenAI with Valve for DOTA2, DeepMind with Blizzard for StarCraft II), there has not been access for other researchers.

**Performant and scalable benchmark**. A single research machine can run experiments in a reasonable amount of time (< 1 day). With a cluster the benchmark can scale to much larger training. RL research is typically limited by one of these problems: being too slow for a single researcher to experiment on a local machine or being too limited in infrastructure to leverage more resources when needed. This makes the benchmark valuable to a wide range of research applications and practitioners.

**Important open problem**. Despite successes in RL for targeted problems in games, generalizing well across opponents (in the multiagent setting) remains an open challenge. To date no publicly available systems have exposed this problem, limiting research on the topic. The topic is important in games in particular and RL in general.

**Weaknesses:**

**Unclear relevance for action space generalization**. The paper does not articulate why using different action spaces would be relevant outside the games context. Why is it helpful to have robotic agents that can control different arms (the example given in the paper)? Of the two types of generalization exposed, this form seems more niche to the games RL community.

**(currently) Requires Windows**. For better or worse most research in ML favors using linux. The requirement for the simulator to use Windows will be a (minor) impediment to adoption. The forthcoming linux release mentioned will alleviate this, but it is not currently available and has no timeline for release stated.

**Additional Feedback:**

One topic that is important to address is accessibility of the game simulator. The call for papers defines accessibility including:
> "i.e. the data can be found and obtained without a personal request to the PI"
Requesting the game executable requires an online request form (requesting a .edu email), which is very similar to directly making a personal request to the PI. This impacts accessibility, particularly given no description was given on the approval process for these requests.


From the call FAQ: https://neurips.cc/Conferences/2022/CallForDatasetsBenchmarks
> Q: My dataset requires open credentialized access. Can I submit to this track?
> A: This will be possible on the condition that a credentialization is necessary for the public good (e.g. because of ethically sensitive medical data), and that an established credentialization procedure is in place that is 1) open to a large section of the public, 2) provides rapid response and access to the data, and 3) is guaranteed to be maintained for many years. A good example here is PhysioNet Credentialing, where users must first understand how to handle data with human subjects, yet is open to anyone who has learned and agrees with the rules. This should be seen as an exceptional measure, and NOT as a way to limit access to data for other reasons (e.g. to shield data behind a Data Transfer Agreement). Misuse would be grounds for desk rejection. During submission, you can indicate that your dataset involves open credentialized access, in which case the necessity, openness, and efficiency of the credentialization process itself will also be checked.

What is the credentialization process in use?


# Questions

**[Q0]** The examples demonstrate scaling using a cluster. Is there an example of using a cluster to train that users can reuse and adapt? What types of cluster environment / interface are supported?

**[Q1]** Figure 3: How many matches of a fixed duration (ex: the max duration) would occur over the 4.32 million samples? I do not know how to compare simulation sample rate and the relevant question is the rate of obtaining training episodes for RL. This is hard to infer from the sample rate or training convergence curves.

**[Q2]** Figure 5: What defines human-level? BT is hand authored. Is there evidence showing how this maps to human performance? What "human" level is used (Elo, &c)? Presumably human players vary in skill and there is a wide range of levels of performance.


# Feedback
Below are minor comments and suggestions for the text.

Figure 2: Consider using icons or portraits to make it easier to see which hero is used.

Figures 6 and 7: Please add in the missing error bars.
As a minor point, it might help to match the name order to highlight similarities and differences in generalization over two regimes. That said, I also understand the choice to order by decreasing level of generalization.

Figures 12-14: Which hero(s) were used in these plots?
Perhaps the supplement could include bar plots of Figures 15 and 16 as well? It was hard to read the overlapping radar plots.


**Clarity:**

Yes. Features are documented, the API is demonstrated, and the text is readable.

**Correctness:**

Yes. Evaluation shows common baseline RL algorithms (DQN, PPO) fail to generalize either to different opponents or different controlled agents. The experiment design is sound.

The appendix provides further validation data on the specific baseline network provided through ablations.

**Documentation:**

Yes. The codebase provides docker images to replicate the core process described in the document on other Windows machines. The code provides scripts to launch the demonstrated configurations and replicate the experiments in the paper.

The website is thoroughly documented including setup and use. The configurations and other aspects are somewhat hard to understand, but have examples in context.

Hosting, licensing, and maintenance are all by the team (and company) that makes the game. Ethical and responsible are not relevant to this simulator.

One issue is accessibility: obtaining encrypted game executable requires submitting a form online (that requires a .edu email). This modestly impacts accessibility, particularly given no description was given on the approval process for these requests.

**Ethics:**

No.

**Relation To Prior Work:**

Yes. Prior work that is openly accessible is distinguished by a focus on single action spaces compared to the multiple characters in this paper that provide multiple action spaces (in separate episodes).

**Summary And Contributions:**

The paper presents a benchmark for a competitive 1v1 digital game (Honor of Kings) oriented toward testing RL algorithm generalization when facing different opponents and using different characters with differing action spaces. The benchmark provides access to run a performant game simulator and a gym-like reinforcement learning environment API to the simulator. Included are baseline algorithms including human authored rule-based behavior and baseline RL algorithm implementations stemming from prior research on the game. The paper contributes a scalable game simulator that can stress RL agent generalization across action spaces and different opponent types. Initial examples demonstrate the failures of baseline RL techniques (DQN, PPO) to generalize in either sense on this benchmark, showing it is an open problem.

---

### Official Review · Reviewer_AvnT · 2022-07-04
**Efficient RL interface to commercially successful MOBA testbed; concerning accessibility issues.**

**Rating:** 7
**Confidence:** 4
**Clarity:** The paper is well-written.

**Strengths:**

- The paper presents the first open-source RL interface to a commercially successful MOBA game.
- The RL interface is well-designed and easy to follow.
- The authors show detailed experiments on the effects of computational resources
- The authors have conducted preliminary experiments examining the RL agent’s generalizability and explored ways to improve generalization.

**Weaknesses:**

- **Accessibility is a big issue**. While Honor of Kings Arena (the RL interface) is licensed under Apache 2.0, running it requires downloading the "hok gamecore”, which requires users to fill out a questionnaire to request access and sign a non-commercial agreement. This practice may violate the key criteria listed in the dataset and benchmark track submission instructions:

    > “datasets should be available and accessible, i.e. the data can be found and obtained without a personal request to the PI, and any required code should be open source.”

    While I understand the “hok gamecore” can be difficult to open source, I think the authors should consider dropping the access request and the non-commercial agreement, such as the case for PySC2, StarCraft II Learning Environment ([https://github.com/deepmind/pysc2](https://github.com/deepmind/pysc2)).
    - If eventually an access request and signing a non-commercial agreement are required, this requirement should be clearly communicated in the paper, which is not done in Section 3.
- **The documentation and setup seem poorly organized.** See comments below.




**Additional Feedback:**

Questions:

1. > HeroStatePrivate, which includes the specific kill information for all the heroes in the game

    What does “kill information” mean?
2. > triplet form, i.e., the action button, the movement offset over the x-axis and y-axis, the skill offset over the x-axis and y-axis, and the target game unit.

    Isn’t there four items instead of three?
3. > model_output_name is the output name defined by the model.

    What model?
4. Figure 4. What is “hurt per frame”?

Misc issues:

1. Figures 1 and 2 are very small - almost illegible if the paper is printed out. Please use a larger font. Figure 2 on all heroes should be moved to the appendix.
2. “Existing benchmark environments on RL generality is mainly focusing” → “Existing benchmark environments on RL generality are mainly focusing”
3. “In MetaWorld [27], it presents a benchmark” -> “For example, MetaWorld [27] presents a benchmark”.
4. “1. it was the number of CPU other than GPU” → “it was the number of CPUs other than GPUs”
5. “Did you describe the limitations of your work? [Yes] See Section 7.” How is Section 7 describing the limitation of the work?

**Correctness:**

The paper overall looks technically correct. A couple of more detailed comments:

- PPO implementation: overall, the authors' customized PPO implementation looks correct to me. However, recent works suggest PPO’s implementation details could greatly impact the performance of deep RL algorithms (Engstrom et al., 2019; Andrychowicz et al., 2020; Huang et al., 2022). Given there is no benchmark for authors’ PPO compared to prior work, it’s hard to draw conclusive evidence about the author’s claim on dual-clip PPO.
- DQN implementation: the authors cited (Wei et al., 2021) for DQN but 1) Wei et al., (2021) presented another algorithm called AQL, and 2) the DQN paper citation should be (Mnih el al., 2015). It’s unclear which DQN did the authors use.


References:
- Mnih, Volodymyr, Koray Kavukcuoglu, David Silver, Andrei A. Rusu, Joel Veness, Marc G. Bellemare, Alex Graves et al. "Human-level control through deep reinforcement learning." *nature*
 518, no. 7540 (2015): 529-533.
- Wei, Hua, Deheng Ye, Zhao Liu, Hao Wu, Bo Yuan, Qiang Fu, Wei Yang, and Zhenhui Li. "Boosting offline reinforcement learning with residual generative modeling." *arXiv preprint arXiv:2106.10411*
 (2021).
- Engstrom, Logan, Andrew Ilyas, Shibani Santurkar, Dimitris Tsipras, Firdaus Janoos, Larry Rudolph, and Aleksander Madry. "Implementation matters in deep rl: A case study on ppo and trpo." In *International conference on learning representations*
. 2019.
- Andrychowicz, Marcin, Anton Raichuk, Piotr Stańczyk, Manu Orsini, Sertan Girgin, Raphaël Marinier, Leonard Hussenot et al. "What matters for on-policy deep actor-critic methods? a large-scale study." In *International conference on learning representations*
. 2020.
- Huang, Shengyi and Dossa, Rousslan Fernand Julien and Raffin, Antonin and Kanervisto, Anssi and Wang, Weixun, "The 37 Implementation Details of Proximal Policy Optimization”, In *International conference on learning representations Blog Track*
2022

**Documentation:**

The project setup and documentation have several issues - please fix them.

**Only support the now no-longer-maintained python 3.6.**

First, the project requires python 3.6, which has reached the end of the life cycle in the December of 2021 ([https://peps.python.org/pep-0494/#lifespan](https://peps.python.org/pep-0494/#lifespan)). For a better adoption, I suggest the authors embrace at least python 3.7.1+.

**Installation issue**

I tried at least three ways to install hok_env but all failed. The three ways include creating a conda environment following `env.yaml` from the hok folder, creating my own virtual environment, and using different versions of python. I also followed the instruction below

```
git clone hhttps://github.com/tencent-ailab/hok_env.git
cd hok_env/hok_env/
pip install -e .
```

which has a typo - “hhttps://github.com/tencent-ailab/hok_env.git” → “https://github.com/tencent-ailab/hok_env.git”. Nevertheless, I always get the following error.

```python
(hok_env) (miniconda3-4.7.12) ➜  unit_test git:(master) ✗ python test_env.py
Traceback (most recent call last):
  File "test_env.py", line 5, in <module>
    from hok import HoK1v1
  File "hok_env/hok_env/hok/__init__.py", line 12, in <module>
    from hok.env1v1 import HoK1v1
  File "hok_env/hok_env/hok/env1v1.py", line 21, in <module>
    import hok.lib.interface as interface
ImportError: hok_env/hok_env/hok/lib/interface.so: invalid ELF header
```

Please consider using a package manager like poetry and pipenv to resolve and lock dependencies. The current list of dependencies is not consistent. For example, the `env.yaml` depends on `numpy==1.19.5, opencv-python==4.5.3.56` ([https://github.com/tencent-ailab/hok_env/blob/464f305d42fb6cad2442cc783527d20d5b077c80/hok_env/env.yaml#L28-L29](https://github.com/tencent-ailab/hok_env/blob/464f305d42fb6cad2442cc783527d20d5b077c80/hok_env/env.yaml#L28-L29) ), but `opencv-python (4.5.3.56) depends on numpy (>=1.21.0)`.

**Conflicting Installation Instructions**

In [https://aiarena.tencent.com/hok/doc/quickstart/index.html](https://aiarena.tencent.com/hok/doc/quickstart/index.html) it says “Currently `hok` only supports environment under Linux.” but in the main repo [https://github.com/tencent-ailab/hok_env](https://github.com/tencent-ailab/hok_env)  says Windows 10 /11 is required.

**Missing CI**

There are barely any tests and no CI to ensure at least the basic `test_env.py` would run without error.

**Stale link for downloading game engine**

The paper mentioned that “The encrypted game engine and game replay tools can be downloaded from: [https://aiarena.tencent.com/hok/download](https://aiarena.tencent.com/hok/download)”, but when clicking on the link it shows

```python
<Error>
<Code>AccessDenied</Code>
<Message>Access Denied.</Message>
<Resource>kaiwu-hok-env-public-1258344706.cos.ap-shanghai.myqcloud.com/hok/download</Resource>
<RequestId>NjJjMzZjOWJfNzQzN2YyMDlfNzA1M19hZjNiY2M1</RequestId>
<TraceId>OGVmYzZiMmQzYjA2OWNhODk0NTRkMTBiOWVmMDAxODc0OWRkZjk0ZDM1NmI1M2E2MTRlY2MzZDhmNmI5MWI1OTVmYzc0YmEzZDg5YTVjMDA4YzA3ZWY5ZjQwZTE2ZGU1MTE5N2Q3ZmIwN2M0MWZlZWViZTBmZTIwMzc4M2U4NjU=</TraceId>
</Error>
```


**Relation To Prior Work:**

Yes, the authors clearly discussed how this work differs from previous contributions.

**Summary And Contributions:**


This paper introduces a new RL simulation environment called "Honor of Kings Arena" for training agents to play **1 vs 1** Honor of Kings, a popular multi-player game worldwide. *Honor of Kings Arena* has an efficient game engine, and combined with the authors' training pipeline, researchers can train a viable agent in just 7 hours with 128 CPU cores and (presumably) a GPU.

Further, the authors suggest Honor of Kings Arena can be an excellent testbed for measuring generalization because the agent needs to learn to 1) control different heroes with different skills and 2) play against different heroes. They have conducted a series of experiments, showing that the agents could overfit and lose to the heroes they were not trained against. To mitigate this issue, the authors have experimented with more diverse training heroes and model distillation that showed promising results.

The authors clearly put tremendous effort into open-sourcing this testbed, and this work has great potential. However, I can't give out a high rating given the accessibility and documentation issues described below.

---

### Official Review · Reviewer_9D3M · 2022-07-27
**Good implementation but unclear relevance**

**Rating:** 6
**Confidence:** 2
**Clarity:** The paper is clear and well-written.

**Strengths:**

- Speed and accessibility of implementation.
- Existing interest in this game environment from the community.
- Comparison of direct transfer learning, multi-task, and distillation setups was interesting.

**Weaknesses:**

- The generalization framing is not very well supported, motivated, or contextualized with regards to prior works. It's not really explained or justified why the type of variation captured by this benchmark (aka certain gameplay elements via the choice of target and opponent) is significant or interesting compared to existing benchmarks.
- It is unclear if this is a challenging enough benchmark to drive forward research progress, given the performance of the baselines.

**Additional Feedback:**

- In Figures 6/7, it would be helpful to highlight the 5 other opponents/targets used for training in the multi-task setting (it's hard to visually map them to the text in the figure captions).

**Correctness:**

The evaluation methods and experiments seem the be constructed and performed correctly.

**Documentation:**

The benchmark is very well documented in the code as well as the supplementary materials.

**Ethics:**

No ethical concerns.

**Relation To Prior Work:**

The selection of prior works discussed feels a bit haphazard and incomplete, missing several relevant RL generalization benchmarks (e.g. NetHack, XLand). I would recommend looking through some of the benchmarks listed in "A Survey of Generalisation in Deep Reinforcement Learning" by Kirk et al. and expanding the related work section.

**Summary And Contributions:**

This paper presents a benchmark based on the game Honor of Kings, where a player can take on one of many different roles and compete against a similarly wide selection of opponents. The motivation is that generalization across these two axes (targets and opponents) is equivalent to generalization across several game dynamics including action control.

The key contributions are:
- An optimized and simplified 1v1 Honor of Kings game engine and interface for RL research.
- A series of baselines and experiments.

---

### Official Review · Reviewer_WLzS · 2022-07-27
**A good environment for evaluating generalization ability and is fully validated**

**Rating:** 8
**Confidence:** 4

**Strengths:**

The Honor of Kings Arena environment focuses on generalization ability evaluation in competitive games, which is a research hotspot in the reinforcement learning domain. The authors conducted several experiments to show that the environment is of high performance, and is feasible for comparing the generalization ability between different agents. The engine is accessible and the APIs are well-designed. RL research related to generalization is expected to benefit from the environment.

**Weaknesses:**

The generalization challenges of Honor of Kings Arena come from different opponents and different targets. Actually, more challenges can be manufactured in this environment, simply by introducing various goals or small tasks, such as "killing the opponent hero as fast as possible", "gaining as much money as possible", or "cooperating with other heroes on some small tasks". Different goals can make the agents more generalizable, and small tasks can lower the bar for training and learning.


**Additional Feedback:**

Some typos:

Line 181: there are two "the"s at the end of the line.

Line 248: policy distillation that "are" proposed in [6], should be "is".

**Clarity:**

The paper is well-structured and easy to read.


**Correctness:**

The experiment design is appropriate and performed correctly. The RL methods (i.e. PPO and DQN) used in the experiments are not SOTA, however, as the paper is submitted to the Datasets and Benchmarks Track, the minor weakness is acceptable.

**Documentation:**

Both the engine and agents are published online. Parameters used in training and evaluation can be found in the attachments or the codes.


**Ethics:**

There is no ethical concerns.

**Relation To Prior Work:**

The authors have discussed several related environments in the Motivations and Related Work Section.

**Summary And Contributions:**

The paper introduces an environment called Honor of Kings Arena, which is derived from the popular mobile game Honor of Kings. The environment can be used for evaluating the generalization ability of agents in competitive games. The authors publish the environment engine and provide easy-to-use interfaces along with detailed specifications. By comparing the performance of two different RL-based methods (i.e. PPO and DQN) and one rule-based method (i.e. BT), the results show that the environment is efficient for training learning algorithms. Finally, the paper demonstrates that the environment imposes generalization challenges across opponents and targets on the agents.

---

### Meta-Review · Area_Chair_jDqN · 2022-09-06

**Recommendation:** Accept
**Confidence:** 4

**Metareview:**

This paper introduces a novel RL benchmark based on the popular computer game "Honor of Kings Arena".
There are two modes of evaluation, one is a single agent task of beating the computer AI and the second one is a competitive setting (two player zero-sum).

The game offers interesting challenges from a generalisation and transfer point of view and will make a valuable contribution to the research landscape.
The authors have included basis baselines in the evaluation.

One suggestion I have is to implement the training on the GPU to reduce computational overhead for scientists.

---

### Decision · Program_Chairs · 2022-09-16

Accept